# Who Are Dispensed the Bulk Amount of Prescription Opioids?

**DOI:** 10.3390/jcm8030293

**Published:** 2019-03-01

**Authors:** M. Mofizul Islam, Dennis Wollersheim

**Affiliations:** 1Department of Public Health, La Trobe University, Melbourne, Victoria 3086, Australia; 2Health Information Management, School of Psychology and Public Health, La Trobe University, Melbourne, Victoria 3086, Australia; D.Wollersheim@latrobe.edu.au

**Keywords:** opioid, prescription opioid, dispensing, chronic user, Australia, defined daily dose

## Abstract

Background: Excessive and non-medical use of prescription opioids is a public health crisis in many settings. This study examined the distribution of user types based on duration of use, trends in and associated factors of dispensing of prescription opioids in New South Wales and Victoria, Australia. Methods: 10% sample of unit-record data of four-year dispensing of prescription opioids was analysed. Quantities dispensed were computed in defined daily dose (DDD). Multilevel models examined factors associated with the duration of dispensing and the quantity dispensed in local government areas. Results: Overall, 53% were single-quarter, 37.3% medium-episodic (dispensed 2–6 quarters), 5% long-episodic (dispensed 7–11 quarters) and 5% were chronic users (dispensed 12–14 quarters). More than 80% of opioids in terms of DDD/1000 people/day were dispensed to long-episodic and chronic users. Codeine and oxycodone were most popular items—both in terms of number of users and quantity dispensed. Duration of dispensing was significantly higher for women than men. Dispensing quantity and duration increased with increasing age and residence in relatively poor neighborhoods. Conclusions: Although only 5% were chronic users, almost 60% of opioids (in DDD/1000 people/day) were dispensed to them. Given that chronic use is linked to adverse health outcomes, and there is a progression toward chronic use, tailored interventions are required for each type of users.

## 1. Introduction

In many countries there has been a growing concern about the increase in utilisation of prescription opioids (PROP) such as tramadol, oxycodone, fentanyl, codeine, morphine and buprenorphine. Prescription opioids are agonists at the mu-opioid receptor. They produce analgesic effect and some adverse effects including dependence, addiction and constipation [1,2,3]. Excessive and or non-medical utilisation of these medicines are now a public health crisis in many settings, with notable impacts recorded in the USA [4], Canada [5] and Australia [6]. While opioid analgesics rapidly relieve many types of pain and improve function, the evidence that there is a positive correlation—much of which is causal—between the magnitudes of PROP dispensing and adverse health outcomes arising from both their medical (e.g., dependence) and nonmedical use (e.g., fatal overdoses) [7,8] is growing. Thus, a long-term treatment using PROP to treat chronic non-cancer pain remains controversial [9]. Although of lesser magnitude than the USA, Australia is also feeling pain from this crisis. The combination of poor health outcomes from long-term utilisation, [10] inappropriate prescribing for pain, and non-medical use of opioids has resulted in a range of adverse consequences including high number of opioid-related deaths [11,12].

An effective public health response to opioid crisis requires understanding of the distribution of, and the temporal and spatial trends of PROP utilisation. In the United States, a recent study of pharmacy data for privately insured adults without cancer found that a small proportion (≈10 percent) of opioid users account for the vast majority of opioids use [13]. However, little is known about the magnitude and trends in short-term, episodic or chronic users of these medications in Australia. Recent dispensing data suggests although overall utilisation of opioids remains mostly stable, the number of individuals who were dispensed an opioid increased over the years [14]. In these circumstances, it would be valuable to know if the growth in number of individuals is attributable to short-term or chronic users. This study, therefore, examined the distribution of PROP user types in New South Wales and Victoria, the two most populous Australian states, trends in PROP dispensing and the associated factors.

## 2. Experimental Section

### 2.1. Prescription Opioid Dispensing in Australia

Based on subsidy type, all prescription opioids that are dispensed from the community pharmacists in Australia can be classified in four categories, namely Pharmaceutical Benefits Schemes (PBS), Repatriation Pharmaceutical Benefits Scheme (RPBS), under co-payment and private. Under Australia’s universal healthcare scheme, which is also known as Medicare, around 80% of all prescription medicines dispensed are subsidized by the PBS [15]. The RPBS is for military veterans, funded by the Department of Veterans’ Affairs. There is also a safety net threshold on PBS items. When threshold is reached, general patients are entitled to PBS medications at the concession price; and concessional patients are entitled to PBS medications at no cost—for the rest of that calendar year. The price of some PBS items is below the general patient co-payment, and thus patients pay the full cost for those items. Some medicines are not listed in the PBS or RPBS, and thus patients pay the full price for those items as a private prescription.

### 2.2. Dataset

Ten percent sample of de-identified unit-record dispensing data of prescription opioids was supplied by the Australian Government Department of Human Services. The dataset was extracted based on the date of supply. In the database, medicines are recorded according to the World Health Organization (WHO) Anatomical and Therapeutic Chemical classification [16] and with information about three types of subsidy schemes. Prescription opioids that were dispensed through private prescription were not included in this dataset. The dataset also contained information about users’ sex (male and female); age in years; date-month-year of dispensing; generic name, form and strength of opioids; the quantity dispensed; and the local government area (LGA) in which the medicines were dispensed. 

Population data for individual states and LGAs, and Socio-Economic Indexes for Areas (SEIFA) were obtained from the Australian Bureau of Statistics. SEIFA data are derived from a set of information collected in five-yearly national census. SEIFA ranks areas according to relative socio-economic disadvantage, higher scores on the Index of Relative Socio-economic Disadvantage indicate a lower level of disadvantage and lower scores indicate a higher level of disadvantage. LGA urbanization (i.e., urban or rural) level was determined by the Australian Classification of Local Government, as categorized in 2013 [17].

### 2.3. Outcome and Exposure Variables for Regression Models

There were three outcome variables, *namely* the duration (in terms of number of quarters) in which the participants were dispensed an opioid, quantities dispensed per prescription in terms of defined daily dose (DDD) units, and age-sex adjusted quantities of opioids in terms of DDD/1000 people/day. The DDD unit was introduced by the World Health Organization Collaborating Centre for Drug Statistics Methodology [16] and corresponds to an estimated mean daily dose of the medicine for an adult when used for its main indication. Using this DDD factor we computed the quantities dispensed per prescription per person as being ∑(*Total units dispensed per prescription* × *weight of each unit ÷ DDD factor for the respective items*) ÷ *Total number of prescriptions dispensed during the study period*. Finally, we computed DDD/1000 people/day for individual opioid types, LGAs, age-groups, sex and four types of SEIFA. Details about this computation could be found elsewhere [18]. Following the direct standardization approach [19] the quantities dispensed in terms of DDD/1000 people/day for individual LGAs were adjusted for 2016 population structure of respective LGAs. The adjusted DDD/1000 people/day thus offers an overall estimate of the prevalence of dispensing in the respective LGAs. Exposure variables were age-group, sex, year, urbanization (rural *vs*. urban), state, four levels of SEIFA, LGAs, and prevalence of cancer in LGA levels.

### 2.4. Analysis

This study analysed four-year (1 January 2013 to 31 December 2016) dispensing data. Based on dispensing dates we grouped the users in 16 three-monthly quarters. A person was coded a user if he/she was dispensed at least once during that quarter. This way each person was coded as a user for at least one and maximum of 16 quarters. All users were then categorized into following four types: single-quarter user (if dispensed only during one quarter); medium-episodic user (if dispensed 2–6 quarters); long-episodic user (if dispensed 7–11 quarters); and chronic user (if dispensed 12–16 quarters).

Descriptive statistics were used to examine the basic features of the users, and quantity dispensed in two states, across major age-groups and among men and women. The quantity of each drug dispensed was estimated in DDD/1000 people/day for both individual states and years.

We developed two regression models. The first model identified relationship between the *number of quarters opioids were dispensed* and the associated factors such as age, sex, location etc. The distribution of number of quarters opioids were dispensed to the users was similar to count data. As there was a hierarchical structure (e.g., LGAs are nested in states) in the dataset, we performed likelihood test to compare random effects model against fixed effects model. Statistically significant results (*p* < 0.05) in this test implied that the random effect models were preferable for modelling this data. Accordingly, we conducted multilevel mixed effects negative binomial regression using *menbreg* commands of STATA program [20]. As the dataset is large and the multilevel model needs calculation of residuals at each iteration and hence a powerful computer with large memory, we took a random sample of 10,000 patients for developing this model. 

The second model examined the relationship between LGA level dispensing of opioids in terms of DDD/1000 people/day (adjusted for population structure of the respective LGAs) and the associated factors. The distribution of LGA level dispensing in terms of DDD/1000 people/day was skewed and needed square root transformation. For this model we conducted multilevel mixed effects generalized linear regression using *xtmixed* commands of STATA program [21].

The total quantity of opioids dispensed in DDD/1000 people/day (adjusted for age and sex) in individual LGAs, and among the chronic users were grouped into quartiles. LGAs that were in the fourth quartile in terms of overall dispensing in DDD/1000 people/day in individual years were identified and mapped. R software version 3.4.4 [22] was used for mapping and the *tidyverse* and *tmap* packages [23,24] to generate the maps.

### 2.5. Ethics Approval

The study has been approved by the Ethics Committee of the La Trobe University (reference number S17-003). 

## 3. Results

Overall, more than half (52.7%) were single-quarter users, 37.3% were medium-episodic users (used 2–6 quarters), 5% were long-episodic users (used 7–11 quarters) and 5% were chronic users (used 12–16 quarters) (Figure 1). More women (53.5%) used opioid than men (46.5%). The mean age of single-quarter, medium-episodic, long-episodic and chronic users were 49 years, 59 years, 65 years and 64 years, respectively. The mean ages for all four categories of user-types were higher (range 1–6 years) for women than men. Women were dominant among all four types of users and the gaps between women and men increased gradually from single-quarter to chronic users. This trend for proportions was statistically significant (*p* < 0.001).

The single quarter and medium-episodic users were predominantly of 20–44 years age-group. On the other hand, the regular and long-episodic users were predominantly the senior citizens (age 65+ years). The proportions of long-episodic and chronic use gradually increased with increasing age of the users. 

The trends in utilisation across the age-groups were similar for both New South Wales and Victoria, although the mean age for all four types of users was 1–2 years higher among users of New South Wales than that of Victoria. 

Codeine and derivatives were the most popular item in terms of number of users being dispensed during the study period, followed by oxycodone (and derivatives) and tramadol. This observation was consistent also in terms of DDD/1000 people/day, for both states, men and women. Item-wise average of four-year dispensing in terms of DDD/1000 people/day were also similar in both states, with little higher dispensing in New South Wales than in Victoria for oxycodone and derivatives and fentanyl. 

The mean quantity per prescription per year for chronic users was 13.5, long-episodic users was 8.3, medium-episodic users was 5.6 and single-quarter users was 4.5 DDD. The mean quantity in DDD per prescription for very high SEIFA was 8.4, high SEIA was 10.2, moderate SEIFA was 10.3 and low SEIFA was 10.5. Quantities per prescription dispensed were gradually increasing with the duration of dispensing. For instance, people who were dispensed for 5 quarters received less opioids per prescription than the people who were dispensed for 6 quarters. Year-wise data of the quantity dispensed per prescription are shown in Figure 2. Quantity per prescription dispensed slightly decreased over the years for all four user types except for long-episodic users in 2016. 

In four years (2013–2016) an average 15 DDD/1000 people/day were dispensed in both New South Wales and Victoria. Around two-thirds (66%) of this quantity were dispensed to the chronic users followed by long-episodic users (15%). Codeine (and derivates), oxycodone (and derivatives) and tramadol were the most utilised items in terms of DDD quantity. The overall quantities dispensed in terms of DDD/1000 people/day were similar for both New South Wales and Victoria over the years.

The incidence rate ratio (IRR) for the number of quarters being dispensed was significantly higher for women than men—meaning that the women were more likely to have dispensed in longer duration than men (Table 1). There was an age gradient with IRR, which indicates an increasing trend in longer duration of dispensing with increasing age. Similarly, compared to people leaving in the very high advantaged SEIFA areas those who were living in high, moderate or least advantaged areas were significantly more likely to have been dispensed a longer duration. 

The average amount of opioids dispensed per LGA was 18.92 DDD/1000 people/day (95% CI: 18.31–19.54; SE ± 0.31). The quantity dispensed in LGAs in terms of DDD/1000 people/day varied from 2.79 to 73.23. As per the regression model with LGA level DDD/1000 people/day, adjusted for age and sex, the likelihood of quantity of opioids dispensed for LGAs decreased with increasing prevalence of cancer. In this model SEIFA was found to be a significant factor, and compared to very high SEIFA (high advantaged areas), people who were living in high, moderate or least SEIFA areas were likely to use higher amount of opioids in terms of DDD/1000 people/day (Table 2). 

LGAs that were in fourth quartile (*n* = 74) in terms of DDD/1000 people/day dispensed were depicted in Figure 3. Of these 74 LGAs, 54 were in New South Wales and the remaining 20 were in Victoria; 40 were in fourth quartile in all four years, 11 in three years, 12 in two and remaining 11 were in only one year. Similarly, in terms of DDD/1000 people/day dispensed for the chronic users 71 LGAs were in the fourth quartile. Of these, 50 were located in New South Wales and the remaining 21 were from Victoria; 41 were in fourth quartile in all four years, 12 in three years, 10 in two and remaining 8 were in only one year. 

## 4. Discussion

Our overall findings suggest that although more than half of the users were dispensed opioids only for a single-quarter and that only five percent were chronic users (used 12 or more quarters), almost 60% of all quantities were dispensed to chronic users. Our findings also suggest that the quantity dispensing per prescription slightly decreased over the years for most categories of users. However, opioid dispensing was relatively high in socioeconomically low-disadvantaged areas. The trends in continuing use of prescription opioids, and increasingly in higher quantity per quarters are concerning because the evidence surrounding the efficacy of opioid treatment in longer term is lacking [9]. Rather, there is growing evidence of risk of serious harms. Prolonged utilisation to opioids result in tolerance, reduced efficacy of opioids and increasing dose [25]. Literature also suggests high opioid dose is associated with an increased risk of opioid-related harm including overdose and death [26,27]. There is also risk of iatrogenic dependence [28]. 

The distribution of users over the 16 quarters demonstrates that more than half of the users were dispensed for single-quarter, and the duration decreased sharply. However, in terms of DDD/1000 people/day around 80% of the quantity dispensed was for the chronic and long-episodic users. Together these observations suggest bulk amount of opioids was used mostly by a small subset of people. Given that a prolonged opioid use is potentially harmful for health and wellbeing of the users, the interventions to curb excessive dispensing of opioids should pay specific attention towards the people who use for a longer duration. However, this in no way should undermine the importance of reducing the number of first-time or medium-episodic users, as the individuals with longer initial prescription periods are at risk of becoming long-episodic or chronic users [29]. A recent report from the Centers for Disease Control found that when the first episode of opioid use lasts for eight or more days, the probability of using those drugs for more than a year increased to 13.5 percent. Patients initiated on long-acting opioids had the highest probabilities of long-term use [29]. Perhaps, tailored interventions are required for each type of these users.

SEIFA is significantly and positively associated with the duration in which opioids were dispensed and with the LGA-level dispensing of opioids in terms of DDD/1000 person/day. Together these observations suggest people living in relatively high disadvantaged areas were not only more likely to use opioids for a longer duration but also to use in larger amount. Also, it could partly be due to higher proportions of population in socioeconomically disadvantaged areas (Low SEIFA Areas) were dispensed opioids than in advantaged areas. Although the underlying reasons for dispensing for a longer duration (and to more people) in relatively disadvantaged areas are beyond the scope of this study, multiple reasons may explain this, such as lack of alternative treatment in socio-economically disadvantaged areas, high prevalence and incidence of chronic illnesses or pain [12], high concentration of concessional patients [30], and or social characteristics such as residents’ emotional distress, low social support and participation in deprived areas [31]. Further studies are needed to unfold as to what extent and how these factors impact on dispensing or utilisation of opioids. 

This study found a substantial geographical variation in dispensing of prescription opioids in terms of adjusted DDD/1000 people/day. While a higher level of dispensing does not necessarily indicate an excessive level of utilisation, this however warrants further and area-based research [32]. Particularly, it might be valuable to know the factors that drive a low- and a high-level dispensing. Having a sound knowledge about these may help implement appropriate public health programs. A strong association of SEIFA suggests there are certain “upstream” factors deeply embedded in some locations. Excessive use of opioids is just one of the many negative consequences of poor socioeconomic conditions and inequity. This perhaps reminds us the importance of having long-term structural interventions to ensure a favourable set of social determinants of health [33]. Further research is needed to unfold this. It should be noted here that an area being identified as having certain characteristics have also certain degree of variation within its boundary, and does not equally apply to all individuals living in the area [34].

Our findings suggest sex and age were significantly associated with dispensing duration of opioids. This observation is consistent with those found in previous studies [14,35,36,37]. The association of sex could partly be attributed to a greater propensity among women to report symptoms, seek treatment and live longer than men. Also, there is a greater prevalence and sensitivity of pain among women than men and hence utilisation of pain medications [36,38]. Among the opioid users, the rate of progression to dependence was quicker among women than men [39], women experience more craving for opioids, and higher level of psychological and or emotional distress are were found to be a significant risk factor for excessive use of prescription opioid use [40]. The likely reason for age being a significant factor was a relatively high prevalence of pain among the growing older people [41]. 

The growth in dispensing of opioid analgesics is associated with escalating opioid overdose fatalities, and undue utilisation by many. Currently considerable volume of scientific, clinical and industrial research is directed towards understanding pain, opioid abuse, development of safer and more effective treatments and preventive interventions for pain. For example, development of abuse-deterrent formulations is considered a high public health priority by the Food and Drug Administration, USA [42]. Given that undue and long-term opioid use is potentially harmful for health and wellbeing of the users, it is important that the public health measures such as up-to-date guidelines about prescription and dispensing, alternative pain management and real-time prescription drug monitoring program are implemented. The Victorian government is currently implemented SafeScript, a real-time monitoring program. It was commenced in 2018, initially focussing on the Western Victoria Primary Health Network catchment area, extending to the rest of Victoria in early 2019 [43]. The New South Wales government is waiting for a nation-wide implementation of this program [44]. Necessary efforts should be made to implement such a program in all states and territories with no delay, and with proper planning so that the known concerns [45] can be minimised.

### Strengths and Limitations

Our study has several strengths and limitations. Firstly, 10% sample of PBS, RPBS and under co-payment dispensing, that constitute more than 80% of all dispensing [46] offered a representative population-level and a reliable estimate. Secondly, multilevel modelling improved the precision of our findings. Thirdly, the second multilevel model that predicted LGA-level dispensing was adjusted for the prevalence of cancer in respective LGAs. On the other hand, our study could not capture opioids dispensed during hospital stays. However, the majority of opioids are dispensed in retail pharmacies in Australia. Additionally, dispensing does not equate to consumption and may not reflect the actual level of exposure to prescription opioids. Given that there remains geographic variability in opioid dispensing, our results may not be generalizable to other jurisdictions. In addition, the classification is applicable for the duration of our study only. Some users may have been classified to single-quarter or medium-episodic users but would be classified as chronic or long-episodic users if data for further years had been available. Another limitation is arbitrary categorization of four user types. In fact, in the literature there is no agreed taxonomy about this. For instance, one study categorized episodes of use longer than 180 days as chronic opioid use [47], while another study classified chronic users as patients who filled more than 10 prescriptions or had more than 120 days’ supply in a given year, with the remaining opioid users being defined as intermittent users [48]. Hadlandsmyth et al. defined patients with either a single opioid prescription or two prescriptions separated by more than 90 days as short-term users and, a period of >90 days of continuous opioid use within the calendar year as long-term users [49].

## 5. Conclusions

Single-quarter opioid users constituted more than half of all users and was the dominant group during the study period (2013–2016). However, fewer long-term and chronic users were dispensed more than 80% of opioids. There were large variations in dispensing across the LGAs. A relatively long duration and overall high level of dispensing were observed in LGAs with socio-economically disadvantaged areas. As continuing use of opioids has considerable side effects and excessive use is a key public health concern, area-level research is warranted to understand the underlying factors that drive variation in dispensing. Both short- and long-term programs are recommended. Short-term programs include strict implementation of opioid prescribing and dispensing guidelines, provision of alternative pain management and implementation of prescription drug monitoring program. Long-term programs should aim at structural interventions to eliminate the fundamental causes of economic and social disadvantages and inequality in social determinants of health. 

## Figures and Tables

**Figure 1 jcm-08-00293-f001:**
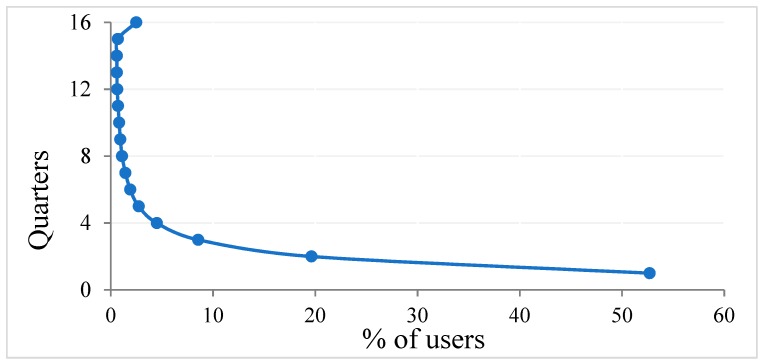
Distribution of users of prescription opioids and the duration of utilisation.

**Figure 2 jcm-08-00293-f002:**
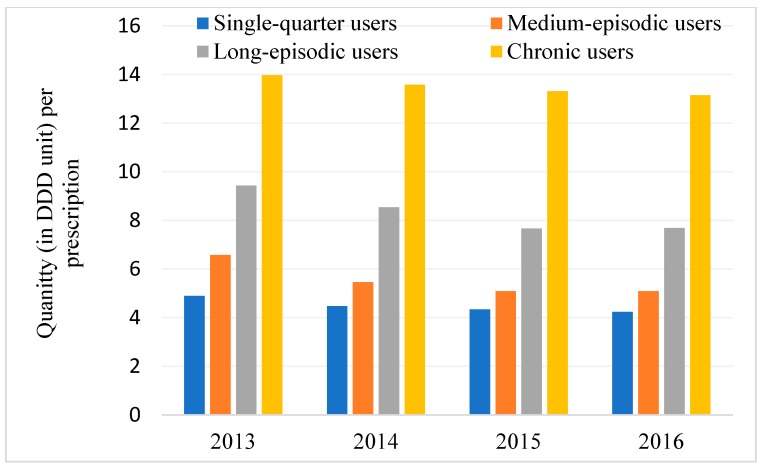
Year-wise trends in dispensing of opioids per prescription in DDD unit.

**Figure 3 jcm-08-00293-f003:**
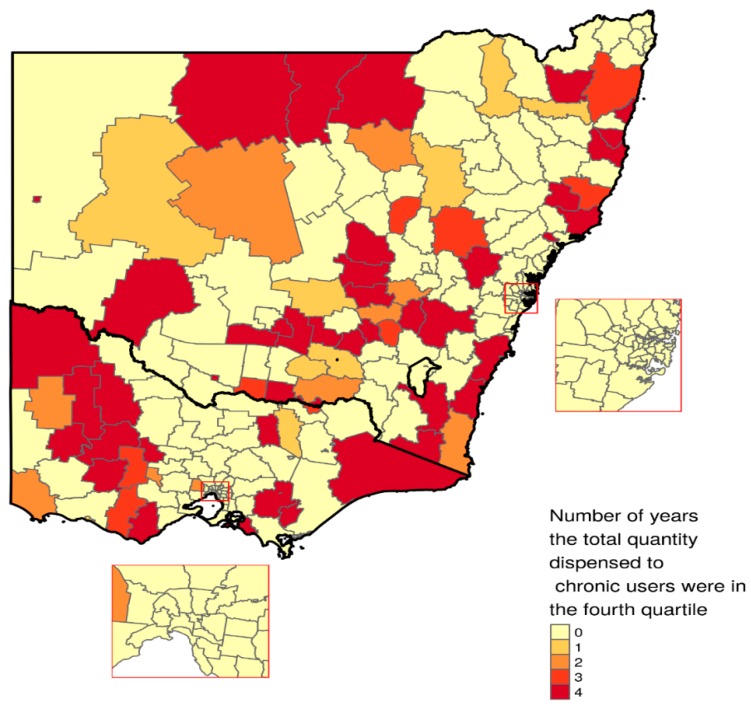
LGAs in New South Wales and Victoria where the total quantity dispensed for the overall population were high (in fourth quartile) in different years (unit: DDD/1000 people/day adjusted for age and sex).

**Table 1 jcm-08-00293-t001:** The relationship between duration of dispensing and covariates in multilevel regression.

Variable	IRR	*p*	95% CI
**Sex**			
Male	1.00	-	-
Female	1.13	<0.01	1.09–1.18
**Age**			
0–19	1.00	-	-
20–44	1.44	<0.01	1.29–1.60
45–64	1.93	<0.01	1.74–2.15
65+	2.69	<0.01	2.43–2.99
**Year**			
2013	1.00	-	-
2014	0.74	<0.01	0.71–0.78
2015	0.63	<0.01	0.60–0.66
2016	0.50	<0.01	0.47–0.53
**SEIFA**			
Very high	1.00	-	-
High	1.17	<0.01	1.10–1.26
Moderate	1.19	<0.01	1.10–1.28
Least	1.22	<0.01	1.13–1.32
**State**			
New South Wales	1.00	-	-
Victoria	1.01	0.76	0.96–1.06
**Urbanization**			
Urban	1.00	-	-
Rural	1.09	0.02	1.01–1.18
Variance (cov.) of random effect		<0.01	
Constant	1.57	<0.01	1.39–1.76
lnalpha	−0.88	-	−0.93 to −0.84
Level 2 (States)	3.03 × 10^−35^	-	-
Level 3 (LGA)	0.01	-	0.01–0.02

Note. IRR—Incidence Rate Ratio; SEIFA—Socio-Economic Indexes for Areas; LGA—Local Government Area.

**Table 2 jcm-08-00293-t002:** Association between LGA-level DDD/1000 people/day and covariates in multilevel regression.

Variable	Coefficient	*p*	95% CI
**Cancer prevalence**	−0.0000243	0.01	−0.001 to −4.5 × 10^−6^
**Year**			
2013 (reference)			
2014	0.03	0.15	−0.01 to 0.07
2015	0.01	0.80	−0.04 to 0.05
2016	−0.09	<0.01	−0.14 to −0.05
**SEIFA**			
Very high (reference)			
High	0.83	<0.01	0.51 to 1.15
Moderate	1.42	<0.01	1.10 to 1.75
Least	1.45	<0.01	1.11 to 1.78
**Urbanization**			
Urban (reference)			
Rural	0.18	0.21	−0.10 to 0.46
Constant	3.34	<0.01	3.06 to 3.62
Random-effects parameters			
States (SD, Constant)	5.58 × 10^−13^	-	2.06 × 10^−26^ to 15.19
LGA (SD, Constant)	0.82	-	0.74 to 0.91
SD (Residual)	0.23	-	0.22 to 0.25

Note: LR test vs. linear model: chi2(2) = 1448.45.

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
