# Peer review of "Who Are Dispensed the Bulk Amount of Prescription Opioids?"

_jcm, 2019, doi:10.3390/jcm8030293_

Round 1
Reviewer 1 Report
The topic of the manuscript is of major relevance in the contemporary context of opioid epidemic. Generally, the manuscript is well written, and appropriate methods have been used to generate data.
There are some specific comments that should be addressed by the authors.
1. Introduction: line 30. It should be exemplified the most prescribed opioids. Also, it is important to mention that prescribed opioids are agonists at the mu-opioid receptor, the primary target for the beneficial (analgesia) as well for non-beneficial actions (addiction, constipation, analgesic tolerance etc).
2. Page 4, line 166. Correct for Figure 2.
3. Figure 3. It should be specified that the chart illustrates data from the New South Wales, Australia.
4. Discussion: Nowadays, there are intensive efforts worldwide (from science, clinicians, industry) directed toward expanding, intensifying, and coordination fundamental, translational, and clinical research with respect to opioid abuse, the understanding of pain, and the discovery and development of safer and more effective treatments and preventive interventions for pain. For example, development of abuse-deterrent formulations (ADFs) is considered a high public health priority by the US FDA. Industry focus on modifying existing opioids to develop ADFs, which intends to make manipulation more difficult or to make abuse of the opioid less attractive or less rewarding. The authors should briefly elaborate on these aspects.
5. Conclusions: line 303. It should read “Single-quarter opioid users”.
Reviewer 2 Report
Good paper, but I suggest the implementation, if it's possible, with single opioids DDD average for each chronic patient.
There is some papers in Literature with this value, useful for a comparison with your data.
Best regards.
